# High Expression of PD-L1 Is Associated with Better Survival in Pancreatic/Periampullary Cancers and Correlates with Epithelial to Mesenchymal Transition

**DOI:** 10.3390/diagnostics11040597

**Published:** 2021-03-26

**Authors:** Nishant Thakur, Kwang Yeol Paik, Gyoyeon Hwang, Yosep Chong

**Affiliations:** 1Department of Hospital Pathology, Yeouido St. Mary’s Hospital, College of Medicine, The Catholic University of Korea, Seoul 07345, Korea; nishantbiotech2014@gmail.com (N.T.); hgy1206@gmail.com (G.H.); 2Department of Surgery, Yeouido St. Mary’s Hospital, College of Medicine, The Catholic University of Korea, Seoul 07345, Korea; kpaik@catholic.ac.kr; 3Department of Hospital Pathology, Uijeongbu St. Mary’s Hospital, College of Medicine, The Catholic University of Korea, 271, Cheonbo-ro, Uijeongbu 11765, Gyeonggi-do, Korea

**Keywords:** programmed death-1, programmed death ligand-1, programmed death ligand-2, recurrence, disease-free survival, overall survival

## Abstract

Periampullary cancers (PACs) are characterized by tumor-infiltrating lymphocytes (TILs), severe fibrosis, and epithelial to mesenchymal transition (EMT). The immune checkpoint marker programmed death-1 (PD-1) and its ligands 1 and 2 have gained popularity in cancers with TILs. Evidence suggests a strong relationship between immune checkpoint markers and EMT in cancers. Here, we evaluated the expression and prognostic significance of immune checkpoint and EMT markers in PAC using an automated image analyzer. Formalin-fixed, paraffin-embedded surgically excised PAC tissues from laboratory archives (1998–2014) were evaluated by immunohistochemical staining for PD-1, PD-L1, and PD-L2 in a tissue microarray. In total, 115 PAC patients (70 males and 45 females) with an average age of 63 years were analyzed. Location, gross type, size, radial resection margin, N-M stage, lymphatic invasion, vascular invasion, perineural invasion, histologically well-differentiated severe inflammation, and high PD-L1 expression were significantly associated with recurrence. Higher PD-L1 expression, but not PD-1 and PD-L2, was significantly related to better overall survival (OS) and disease-free survival (DFS). PD-L1 and PD-L2 were significantly related to EMT markers. Aside from other clinicopathologic parameters, high PD-L1 expression was significantly related to better OS and DFS of PAC patients. Moreover, immune checkpoint markers were significantly associated with EMT markers. Therefore, PD-L1 expression can be a good prognostic marker to guide future immune target-based therapies in PAC patients.

## 1. Introduction

Periampullary cancers (PACs), including pancreatic ductal adenocarcinoma, ampulla of Vater (AOV), and common bile duct (CBD) cancers, are highly lethal malignancies and predicted to be the second most common cause of cancer-related death after lung cancer by 2030 [1,2]. The median five-year overall survival (OS) rate is observed in less than 5% of patients; however, most patients die within a year after diagnosis owing to the invasive and metastatic nature of PACs [1]. Despite advancements in new chemotherapy protocols such as FOLFIRINOX or combination treatment with gemcitabine and nab-paclitaxel [3,4], the survival rate of patients is poor for several reasons. The tumor microenvironment is characterized by profound desmoplasia, epithelial-mesenchymal transition (EMT), cancer stem cells, and immunosuppressive cell infiltrates, which seem to contribute to chemotherapeutic resistance [5,6]. To date, there is no validated prognostic biomarker for therapeutic response; therefore, it is essential to identify a specific novel predictive biomarker that could help in clinical decision-making and the development of better therapies for the management of PAC.

The immune checkpoint marker programmed death-1 (PD-1) and its ligands programmed death ligand-1 (PD-L1) and programmed death ligand-2 (PD-L2) have attracted interest in the field of cancer immunology owing to conflicting prognostic significance [7]. The blockade of this pathway using specific inhibitors such as pembrolizumab and nivolumab could enhance the cytotoxicity of T cells in the tumor environment and substantially increase the long-term survival in different cancers [7,8,9]. However, in colon cancer and PAC, only patients with microsatellite instability-high (MSI-H) and mismatch repair deficiency can benefit from this treatment [10,11,12]. Furthermore, the prognostic significance of PD-L1 has been associated with poor survival in many solid cancers [13]. The results are conflicting in pancreatic cancer, as is evident from the association of PD-L1 overexpression with poor OS in one study [14] and better OS in another report [15]. We hypothesized that these conflicting results on PD-L1 significance in cancers may be related to inter-observer and intra-observer variability between pathologists and different diagnostic assays, including specific antibody clones and staining platform. Nevertheless, the significance of PD-L2 expression has not been well investigated.

Recent evidence suggests a relationship between immune checkpoint markers and EMT. EMT is a multi-step process involving the transition from an epithelial to mesenchymal phenotype in response to chemoresistance and is important for cancer metastasis [16]. Many supporting studies have revealed the link between PD-L1 and the EMT pathway. For instance, Alsuliman et al. demonstrated that EMT upregulated PD-L1 expression through the phosphoinositide 3-kinase/protein kinase B pathway in breast cancer [17]. Another study by Ock et al. showed that PD-L1 expression significantly downregulated E-cadherin and upregulated vimentin expression in head and neck squamous cancer cell lines. Using the Cancer Genome Atlas (TCGA) database, it was shown that patients with positive PD-L1 expression and EMT had a worse prognosis than those with positive PD-L1 and negative EMT [18]. However, to the best of our knowledge, the relationship between immune checkpoint markers and EMT has not been explored in PAC.

In the present study, we used automated image analyzer software to evaluate the expression of PD-1, PD-L1, and PD-L2 and their correlation with EMT markers in a tissue microarray (TMA) section of patients with PAC. We also scrutinized the correlation of these markers with clinicopathological features and determined their prognostic significance.

## 2. Materials and Methods

### 2.1. Patient Enrollment

This study was approved by the Institutional Review Board of the Catholic University of Korea (SC19SESI0112). Surgically excised 115 PAC formalin-fixed paraffin-embedded tissues from Yeouido St. Mary’s Hospital Laboratory Archives, Seoul, Korea, from 1998 to 2014 were obtained. Distal CBD, AOV, and pancreatic head cancers with periampullary involvement were extracted by pylorus-preserving pancreaticoduodenectomy or Whipple surgery, while duodenal cancers without ampullary or pancreatic involvement and intraductal papillary mucinous neoplasms considered as benign or premalignant lesions were excluded. None of the patients had experienced any preoperative chemotherapy treatment. The patients over T stage 3 (>pT3) or nodal metastasis (>pN1) received 5FU or gemcitabine-based adjuvant chemotherapy. A total of 115 patients with an average age of 63 years (range 36 to 82 years), including 70 men and 45 women, were enrolled in the study. Their performance status was checked according to the American Society of Anaesthesiologists (ASA) for operation. None of the patients underwent vascular resection during the surgery. The retrieved cases were blinded using sequential numbering. Y. C. and N. T. independently reviewed hematoxylin and eosin-stained slides to validate the original diagnosis. Cases with insufficient tissue samples were not included in the TMA analysis.

Clinicopathologic information, including age, sex, tumor location, gross type, TNM stage, tumor size (largest diameter), positive radial resection margin, lymphatic, vascular, perineural invasion, histologic differentiation (pancreaticobiliary vs. intestinal subtype, and well, moderately, poorly for each), degree of fibrosis and inflammation (mild, moderate, and severe for each), date of surgery, tumor recurrence, and date of death, were recorded. The gross type was classified based on the morphologic growth patterns of the tumors on the gross examination as fungating/polypoid, sessile, ulceroinfiltrative, and solid subtypes. The degree of fibrosis was explained based on the four-tier system comprising < 10% (none), 10–33% (mild), 34–66% (moderate), and 67–100% (severe). The tumor epicenter was examined according to the staging system of the AJCC/Union International Contre le Cancer staging system 8th edition (AJCC/UICC).

Immunohistochemical staining for CK7, CK20, and CDX-2 was performed to determine pancreaticobiliary and intestinal subtypes. Radiological techniques such as position emission tomography and computed tomography (PET-CT) were performed and used along with changes in serum CA19–9 level to evaluate tumor relapse in patients with surgically resectable cancer. Incompletely resected cases with increasing tumor load were excluded from tumor recurrence, and death resulting from bile leakage bleeding, pulmonary embolism, and infection were not considered as cancer-related deaths. Patients with unspecified reasons for death were excluded from survival analysis. Data regarding the reason and date of death were collected based on the National Death Certificate data and therapeutic records from our institute. As National Death Certificate records have a year of postponement for information assortment, the death data of recent cases in 2013 and 2014 were documented based on our institutional medical records.

### 2.2. TMA Analysis

A Quick-Ray^®^ Tissue Microarray recipient block (UB06–2, UNITMA Co., LTD., Seoul, Korea) was used to prepare nine TMA recipient blocks. Each case represented as three 2 mm sized tumor spots obtained from donor blocks to avoid tissue loss and edge artifact. Each recipient block comprised 45 cores of tumor tissues (15 cases), 4 cores of positive controls for stem cell markers, and 1 core of negative control. The recipient blocks were incubated at 30 °C for 25 min to reduce tissue loss before core insertion. Normal umbilical cord for FGFR1, normal blood vessel for VEGF, and normal liver tissue for IGF1 were used as positive controls.

### 2.3. Immunohistochemistry

The IHC procedure was similar to that used in our previous study. Formalin-fixed and paraffin-embedded TMA blocks were cut into 4 μm thick sections, dried in an oven at 70 °C for 60 min, deparaffinized in xylene, and rehydrated in ethanol. Next, endogenous peroxidase activity was blocked by 3% hydrogen peroxide for 15 min, and antigen retrieval was performed by heating at 70 °C for 1 h, followed by pre-treatment with cell conditioner 2 (pH 6) for 60 min. After 1 h of blocking with 5% normal goat serum, samples were incubated with respective antibodies at room temperature for 32 min, and then stained for 4 min with hematoxylin and the bluing agent. Samples were then subjected to chromogenic detection using the ultraview universal DAB Detection Kit for 2 min and a final washing step for 3 min. Balsamic acid was used to cover stained slides. Immunohistochemical staining for CK7, CDX-2, and CK20 was performed to classify tumors into pancreaticobiliary and intestinal subtypes. In addition, immunohistochemical staining was performed for immune cell markers such as PD-1, PD-L1, and PD-L2 and epithelial-mesenchymal transition markers, including IGF1, FGFR1, and VEGF.

The dilution and incubation conditions for each antibody were as follows: PD-1 (1:100, Cell Marque, CA, USA), PD-L1 (1:1000, Cell Marque, CA, USA), PD-L2 (1:1000, Cell Marque, CA, USA), CK7 (Prediluted, Ventana, Roche Diagnostics, USA), CDX-2 (Prediluted, Ventana, Roche Diagnostics, USA), CK20 (Prediluted, Ventana, Roche Diagnostics, USA), IGF1 (1:100, Abcam, CB4, UK), FGFR1 (ChIP Grade ab10646, 1:100, Abcam, CB4, UK), and VEGF (1:50, Quartett Immunodiagnostika, Berlin, Germany). A three-tier system (0, 1+, 2+, and 3+) was used to evaluate the immunoreactivity intensity score for each TMA core. If the intensity was high or low, the most frequently reported intensity was scored.

### 2.4. Image Analysis

We acquired 4–10 representative images of PD-1, PD-L1, and PD-L2 from the most strongly stained areas for each case and analyzed them using the US FDA-cleared image analysis system, GenAsis^TM^ HiPath (Applied Spectral Imaging Ltd., USA) as per the manufacturer’s instructions. The Korean FDA passed the Annual Quality Assurance program of the Korean Society of Pathologists for Image Analysis for Ki-67. Auto-white balancing was performed before capturing the image. Total and positive cells were counted using a three-tier classification system based on staining intensity using the default settings of the software (negative, 1+, 2+, and 3+; Figure 1). No manual pre- or post-processing adjustments were carried out. 

### 2.5. PD-1, PD-L1, and PD-L2 Counting System

The following immunoreactivity scoring system is summarized in Table 1. For PD-1, one scoring system was used as follows: 0, 1*p* + 2*p* + 3*p* < 1%; 1, 1*p* + 2*p* + 3*p* < 10%; 2, 1*p* + 2*p* + 3*p* ≥ 10%. Two scoring systems were used for PD-L1, and each type was further divided into three classes as follows: PD-L1 score 1 (1: 2*p* + 3*p* < 50%; 2: 2*p* + 3*p* < 90%; 3: 2*p* + 3*p* ≥ 90%); PD-L1 score 2 (1: 1*p* + 2*p* + 3*p* < 90%; 2: 1*p* + 2*p* + 3*p* < 99%; 3: 1*p* + 2*p* + 3*p* ≥ 99%). Likewise, two scoring systems were used for PD-L2. Each type was further divided into four subtypes as follows: PD-L2 score 1 (0: 2*p* + 3*p* < 1%, 1: 2*p* + 3*p* < 10%, 2: 2*p* + 3*p* < 50%, 3: 2*p* + 3*p* ≥ 50%) and PD-L2 score 2 (0: 1*p* + 2*p* + 3*p* < 1%, 1: 1*p* + 2*p* + 3*p* < 10%, 2: 1*p* + 2*p* + 3*p* < 50%, 3: 1*p* + 2*p* + 3*p* ≥ 50%) (Table 1).

### 2.6. Statistical Analysis

R language version 3.3.3 (R Foundation for Statistical Computing, Vienna, Austria) and T&F program version 3.0 (YooJin BioSoft, Korea) were used for all statistical analyses. Data are expressed as sample numbers and percentages. When subtypes of markers were regarded as ordinal variables, Spearman correlation analysis was performed to reveal the linear relationship between markers. In addition, the association between subtypes of markers was analyzed using the Fisher’s exact test. Kaplan-Meier survival curve analysis was performed to test OS and DFS; results with *p* < 0.05 were considered statistically significant. Furthermore, Cox proportional hazard regression analysis was performed to analyze the effect of each independent variable, including markers on survival. Death events with OS time and recurrence events with DFS time were used in the survival analysis. To analyze the independent effect of input variables, multivariable analysis was performed using a backward stepwise variable elimination procedure as a variable selection method to minimize Akaike Information Criterion (AIC). A significance level of 0.1 was used in the univariable analysis to select initial input variables for the multivariable analysis.

## 3. Results

### 3.1. Patient Characteristics

The clinicopathologic data of the enrolled patients are summarized in Table 2. The age of the patients ranged from 36 to 82 years (mean age 63 years). Out of 115 patients, 70 were male and 45 were female (M:F = 1.52:1) (Table 1). For ASA status score, 10 patients (8.69%) were found in score 1, 103 patients were in score 2, and only 2 patients were in score 3. In gross type, 20 cases (17.4%) were classified as fungating type, 83 (72.2%) as infiltrative, 3 (2.6%) as ulcerofungating, 4 (3.5%) as sessile, and 2 (1.7%) as solid type. The mean tumor size was 3.2 cm (range 0.6—8.0 cm). All cases were stratified into two groups based on tumor size as follows: 88 cases (76.5%) in the <4.5 cm group and 27 cases (23.5%) in the >4.5 cm group. The N stage was divided into various groups according to the American Joint Cancer Committee (AJCC) staging system as follows: 66 cases (57.4%) were N0, 49 were N1 (42.6%), and none of the cases were found in N2. Moreover, within the M stage, 107 cases (93.7%) were M0 and 8 cases (6.9%) were M1. In addition, we observed lymphatic invasion in 51 cases (44.3%), vascular invasion in 14 cases (12.2%), perineural invasion in 67 cases (57.3%), positive radial resection margin in 8 cases (6.9%), and tumor ulceration in 9 cases (7.8%). Histologic grade revealed 30 cases (26.1%) as well-differentiated tumors, 81 (70.4%) as moderately differentiated tumors, and 4 as (3.5%) poorly differentiated tumors. In histologic subtypes, 30 cases (26.1%) were of the pancreaticobiliary subtype, 54 (46.9%) were prone to pancreaticobiliary subtype, 16 (13.9%) were prone to intestinal subtype, and 14 cases (12.1%) were intestinal subtype. Fibrosis was absent in 4 cases (3.5%) and was considered as mild, moderate, and severe in 20 (17.4%), 64 (55.6%), and 27 (23.5%) cases, respectively. The degree of inflammation was mild in 39 cases (33.9%), moderate in 63 cases (54.8%), and severe in 13 cases (11.3%). Tumor recurrence was observed in 52 cases (45.2%) during an average follow-up of 969.7 days (range, 3–5234 days). The average disease-free survival (DFS) duration was 731.2 days (range, 3–4173 days). Of the 115 patients, seven (67.8%) died during the follow-up.

### 3.2. Immunohistochemical Staining and Immunoreactivity for Immune Checkpoint and EMT Markers

The immunohistochemical staining conditions are summarized in Table 3 and the representative images are shown in Appendix A. During the analysis of immune checkpoint markers, the immunoreactivity of PD-1 was 1+ in 97 cases (85.1%), 2+ in 11 cases (9.6%), and 3+ in 6 cases (5.3%). PD-L1 score 1 staining of 1+ was observed in 10 cases (8.8%), 2+ in 52 cases (45.6%), and 3+ in 52 cases (45.6%), while PD-L1 score 2 was 1+ in 24 cases (21.1%), 2+ in 44 cases (38.6%), and 3+ in 46 cases (40.3%). PD-L2 score 1 staining was negative in 21 cases (18.4%), 1+ in 31 cases (27.2%), 2+ in 39 cases (.34.2%), and 3+ in 23 cases (20.2%), and PD-L2 score 2 immunoreactivity was negative in 4 cases, 1+ in 14 cases (12.3%), 2+ in 34 cases (29.8%), and 3+ in 62 cases (54%). Evaluation of EMT markers showed that insulin-like growth factor 1 (IGF1) staining was negative in 13 cases (11.3%), 1+ in 70 cases (60.9%), 2+ in 26 cases (22.6%), and 3+ in 6 cases (5.2%). Fibroblast growth factor receptor 1 (FGFR1) immunoreactivity was 1+, 2+, and 3+ in 7 (6.1%), 50 (43.5%), and 58 (50.4%) cases, respectively. Vascular endothelial growth factor (VEGF) immunoreactivity was 1+ in 17 cases (14.8%), 2+ in 71 cases (61.7%), and 3+ in 27 cases (23.5%) (Table 3).

### 3.3. Association of Immune Checkpoint Markers with Other Markers

No significant relationship was observed between the PD1 subtype and other stem cell and EMT markers in the Spearman rank correlation analysis as well as Fisher’s exact test (data not shown). PD-L1 score 1 was significantly associated with VEGF (*p* = 0.006) in the Spearman rank correlation analysis, but not in Fisher’s exact test. Furthermore, we failed to observe any direct correlation between PD-L1 score 2 and any markers using both tests. However, IGF1 (*p* < 0.001) and FGFR (*p* < 0.001) were significantly associated with PD-L2 score 1 in the Spearman rank correlation analysis, while IGF1 (*p* = 0.002), FGFR (*p* < 0.001), and VEGF (*p* = 0.010) showed significant correlation with PD-L2 score 1 in the Fisher’s exact test. Further, IGF1 (*p* < 0.001), FGFR (*p* < 0.001), and VEGF (*p* = 0.003) were significantly correlated with PD-L2 score 2 in both tests (data not shown).

### 3.4. Association of Immune Checkpoint Markers with Clinicopathological Parameters Related to the OS of PAC Patients

Kaplan-Meier test results revealed the significant association between OS and clinical parameters such as lower T7 stage (*p* < 0.001), AOV location (*p* < 0.001), sessile and solid Gross type (*p* < 0.001), size less than 4.5 (*p* = 0.029), lower N stage (*p* < 0.001), and lower M stage (*p* < 0.001). Other clinical parameters were not significantly associated with OS (Appendix A). Among the pathological parameters, no lymphatic invasion (*p* < 0.001), no vascular invasion (*p* = 0.003), no perivascular (*p* < 0.001), histological differentiation (*p* = 0.007), histological intestinal subtype (*p* = 0.002), and no fibrosis (*p* = 0.001) were significantly associated with OS (Appendix A). In addition, immunohistochemistry (IHC) markers such as (CK20; 3+; *p* = 0.029), CDX2 (3+; *p* < 0.001), FGFR (3+: *p* < 0.001), and VEGF (3+: *p* = 0.03) were significantly associated with OS (Appendix A). Among immune cell markers, only PD-L1 (score 1:3, *p* = 0.05; score 2:3, *p* = 0.009; and score 2:3, *p* = 0.002), but not PD-1 and PD-L2 (data not shown), were significantly associated with OS (Figure 1A–C).

In the univariate analysis, location (*p* < 0.001), gross type (*p* = 0.013), size (*p* < 0.001), N1 stage (*p* < 0.001), M stage (*p* < 0.001), lymphatic invasion (*p* < 0.001), perineural invasion (*p* < 0.001), vascular invasion (*p* = 0.004), histological differentiation (*p* = 0.01), degree of inflammation (severe vs. mild: *p* = 0.036), histologic subtype (*p* = 0.004), CK7 (3+ vs. 0; *p* = 0.045), CK20 (*p* = 0.039), CDX2 (*p* = 0.002), FGFR (*p* = 0.003), and VEGF (*p* = 0.037) were significantly associated with OS (Appendix A). Among the immune checkpoint markers, only PD-L1 score 2 (*p* = 0.012) showed significant association with OS (Appendix A).

In the multivariate analysis using Cox regression model, location (*p* < 0.001), N stage (*p* = 0.001), M stage (*p* = 0.004), FGFR (*p* = 0.004), and PD-L1 score 2 (3+ vs. 1+or 2+, *p* = 0.019) were independent prognostic factors of OS in patients with PAC (Appendix A).

### 3.5. Association of Immune Checkpoint Markers with DFS in Recurrent Patients

We performed a Kaplan-Meier test and found that only clinical parameters such as location of AOV (*p* = 0.006), size less than 4.5 cm (*p* = 0.045), and lower N stage (*p* < 0.001) were significantly correlated with better DFS (Appendix A). Pathological markers such as no lymphatic invasion (*p* < 0.001), histological well differentiation (*p* < 0.001), and no fibrosis (*p* = 0.005) failed to show any significant association with better DFS (Appendix A). Among IHC markers, CDX2 (3+: *p* = 0.025) and FGFR (3+: *p* = 0.007) were significantly associated with better DFS (Appendix A). Further, PD-L1 (score 2:3, *p* = 0.027 and score 2:3, *p* = 0.009) was associated with better DFS (Figure 1d,e).

Univariate analysis using the Cox regression model revealed the significant relationship between age (*p* = 0.042), location (*p* < 0.008), gross type (infiltrative vs. fungative: *p* = 0.049), size (*p* = 0.009), N1 stage (*p* = 0.001), lymphatic invasion (*p* < 0.001), histologic grade (*p* = 0.003), histologic subtype (intestinal subtype vs. pancreaticobiliary subtype; *p* = 0.031), CDX2 (*p* = 0.035), and FGFR (*p* = 0.01) with better DFS (Appendix A). Among the immune checkpoint markers, only PD-L1 score 2 (*p* = 0.032) was related to better DFS (Appendix A).

We performed multivariable Cox proportional hazard regression analysis and found age (*p* = 0.01), lymphatic invasion (*p* < 0.001), and PD-L1 score 2 (3+ vs. 1+ or 2+; *p* = 0.014) as independent prognostic factors of DFS in patients with PAC (Appendix A).

## 4. Discussion

In our study, we used automated image analyzer software to count PD-1, PD-L1, and PD-L2 and found higher PD-L1 expression to be significantly related to better OS and DFS in patients with PACs. Immune checkpoint markers were significantly associated with EMT markers. We also validated the significant relationship between OS, DFS, and the previously established clinicopathological prognostic markers, pathological prognostic markers, and IHC markers such as CK20 and CDX2, as previously reported [1,19].

In the present study, higher PD-L1 expression was significantly correlated with better OS and DFS in periampullary/pancreatic cancer patients. PD-L1 plays a major role in the maintenance and control of immune surveillance. Most cancer cells progress by avoiding immune surveillance through PD-L1 overexpression [20,21]. Hence, it is natural to discover PD-L1 association with prognosis. However, there are conflicting results on the prognostic significance of PD-L1 in different cancers. Researchers have studied the prognostic significance of PD-L1 in cancers and revealed conflicting results depending on cancer types. For instance, PD-L1 expression is associated with worse OS in multiple solid tumors, including breast cancer [22], lung cancer [23], colorectal cancer [24], cervical cancer [25], and hepatocellular carcinoma [26]. However, the prognostic significance of PD-L1 is inconsistent and conflicting in patients with PAC because higher PD-L1 expression was significantly associated with poor OS (*p* = 0.016) and progression-free survival [14,27] as well as better OS and DFS [15,28].

The difference in the prognostic value of PD-L1 in previous studies may be associated with the use of different antibody clones and IHC interpretation criteria. In terms of variability in antibody clones, the positivity of SP263 (durvalumab) assay was slightly higher than that of 22C3 pharmDx (pembrolizumab) at the center of tumors and invasive margins of 379 TMA gastric cancer [29]. Similarly, PD-L1 staining was investigated in lung cancer cells using four antibody clones, which showed different PD-L1 patterns [30].

These conflicting results might also be from the complexity of the tumor microenvironment where tumor cells, infiltrating lymphocytes, and accompanying macrophages are all admixed. All of these cells can express PD-1, PD-L1, and PD-L2. However, our interpretation criteria, which is mostly a combined positive score, does not properly represent the different expression pattern of each group of cells.

Many researchers and practicing pathologists have been complaining about variable and complex interpretation criteria according to the clones and the possibility of inter- and intra-observer variability in the interpretation of PD-L1. To date, different cut-off values have been used to interpret IHC results across organs based on antibody clones in all pancreatic cancer studies. Moreover, there is not enough evidence that PD-L1 scoring based on a certain clone can substitute the results of other clones. Only one study has reported inter- and intra-observer variability of PD-L1 expression using four clones, SP263, SP142, 22C3, and E1L3N, in head and neck, breast, and urothelial carcinomas. These authors showed a relatively high concordance between three clones. However, the total sample size was very small (less than 30 cases for each cancer) and the comparison was made between only three pathologists, which cannot exclude the possibility of publication bias. Thus, further evidence from more studies with larger samples is imperative [31]. Therefore, we used automated image analyzer software to reduce subjectivity.

We found that immune checkpoint markers were significantly associated with EMT markers. PD-L1 was significantly associated with VEGF, while PD-L2 expression was related to IGF, FGFR, and VEGF markers. EMT is crucial for cancer cell progression, stemness, therapeutic resistance, and tumor immune evasion. Typical EMT markers that have been explored in solid cancers include Snail, ZEB1, E-cadherin, vimentin, N-cadherin, and more recently, FGFR, VEGF, and IGF. Association between expression of immune checkpoint and EMT markers has been reported in several cancers, including breast, lung, and head and neck cancers [32]. In lung cancer, PD-L1 score was positively associated with Snail and vimentin expression [33]. Similarly, higher PD-L1 expression was reported in mesenchymal breast cancer cell lines and associated with increased activation of ZEB1 or Snail [34]. The results of the present study are consistent with these findings [35].

We re-evaluated other previously well-established markers and found the location of AOV, lower T stage, sessile gross type, size less than 4.5 cm, lower N stage, lower M stage, no lymphatic invasion, no vascular invasion, no perineural invasion, histologically well-differentiated, intestinal subtype, and no fibrosis to be significantly associated with OS, consistent with the results of previous studies. We also compared the 7th and 8th edition of T staging based on the AJCC cancer staging system but failed to notice any significant difference. The identification of tumor epicenters is one of the most crucial factors that can help in the characterization of AOV, pancreatic, and CBD cancers. The AJCC 8th edition also relies on various size norms and can vary according to the location of the tumor. According to our study results, CK20 and CDX2 expression may be good prognostic markers and can serve as additional parameters to guide the pathologist during TNM staging.

Our study has a few limitations. First, considering the rarity of this disease, the sample size included in this study was not satisfactory. Future investigations with more samples from other institutions might be warranted to generalize the findings of our results. Second, this was a retrospective study that could simply reveal the relationship between molecular markers. Additional studies using other techniques and markers are needed to scrutinize the detailed mechanism between PD-L1 and EMT pathways in PACs. Third, in this study, we only included cases with enough follow-up time and tried not to include too recent cases because recent advances in patient management, chemotherapy, and supportive quality might affect the overall and disease-free survival of recent cases as cofounding factors other than the PD-L1 expression (the cases from 1998 to 2014 were included). However, this might not represent up-to-date results regarding many aspects. Thus, interpretation with caution is required.

## 5. Conclusions

In this study, we found that higher expression of PD-L1 was significantly related to better OS and DFS of patients with PAC. Moreover, immune checkpoint markers were significantly associated with EMT markers. Therefore, PD-L1 expression can be a good prognostic marker and could guide future immune target-based therapies in patients with PAC.

## Figures and Tables

**Figure 1 diagnostics-11-00597-f001:**
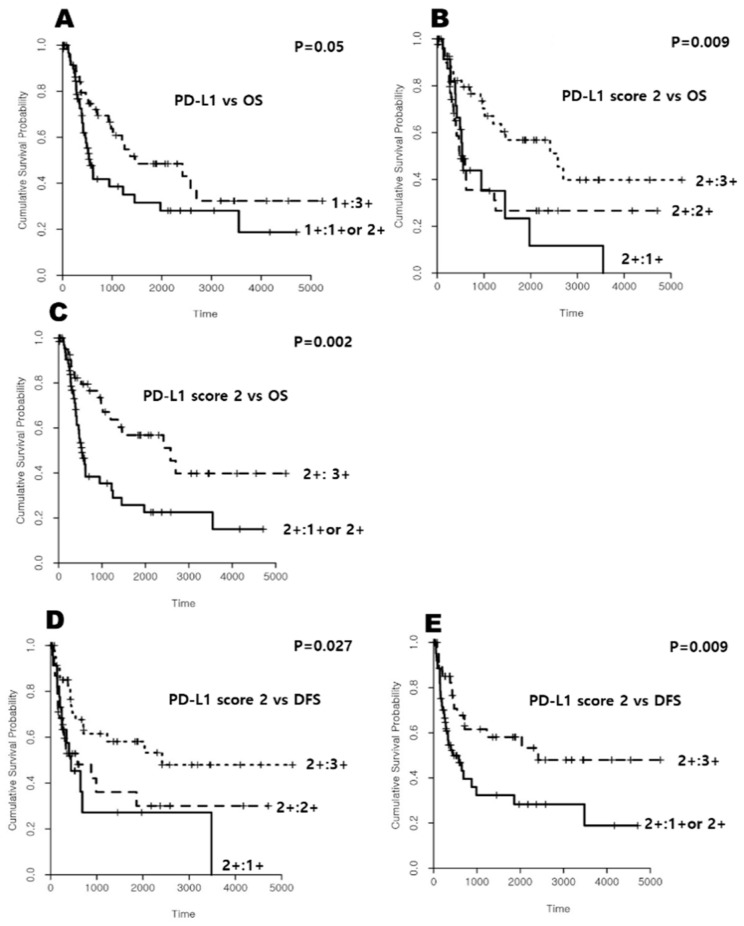
Kaplan-Meier plots on the relationship between immune cell markers and overall survival (OS) and disease-free survival (DFS) in the periampullary/pancreatic cancers patients. There was a significant difference between PD-L1 and OS according to (**A**) PD-L1 (1+:3+ vs. 1+:1+ or 2+), (**B**) PD-L1 (2+:3+ vs. 2+:2+ vs. 2+:1+), and (**C**) PD-L1 (2+:3+ vs. 2+:1+ or 2+), while there was also a significant relationship between PD-L1 and DFS according to (**D**) PD-L1 (2+:3+ vs. 2+:2+ vs. 2+:1+), and (**E**) PD-L1 (2+:3+ vs. 2+:1+ or 2+).

**Table 1 diagnostics-11-00597-t001:** Scoring system for immune checkpoint markers using an automated image analyzer.

Scoring	PD1 Score	PD-L1		PD-L2	
		Score 1	Score 2	Score 1	Score 2
0				2*p* + 3*p* < 1%	1*p* + 2*p* + 3*p* < 1%
1+	1*p* + 2*p* + 3*p* < 1%	2*p* + 3*p* < 50%	1*p* + 2*p* + 3*p* < 90%	1% ≤ 2*p* + 3*p* < 10%	1% ≤ 1*p* + 2*p* + 3*p* < 10%
2+	1% ≤ 1*p* + 2*p* + 3*p* < 10%	50% ≤ 2*p* + 3*p* < 90%	90% ≤ 1*p* + 2*p* + 3*p* < 99%	10% ≤ 2P + 3P < 50%	10% ≤ 1*p* + 2*p* + 3*p* < 50%
3+	1*p* + 2*p* + 3*p* ≥ 10%	2*p* + 3*p* ≥ 90%	1*p* + 2*p* + 3*p* ≥ 99%	2*p* + 3*p* ≥ 50%	1*p* + 2*p* + 3*p* < 50%

**Table 2 diagnostics-11-00597-t002:** Summary of clinicopathological data of the enrolled cases.

Clinicopathological Parameters	No. (%)			
Age (yrs)	Ranged 36–82		Mean 63.0 ± 9.4	
Sex	Male 70, Female 45		M:F = 1.52:1	
ASA status score				
Score 1	10		(8.69%)	
Score 2	103		(89.56%)	
Score 3	2		(1.73%)	
Location (tumor epicenter)				
AOV	32		(27.82%)	
Pancreatic head	38		(33.04%)	
Distal CBD	45		(39.13%)	
T stage	T7		T8	
Tis	1	(0.86%)	1	(0.86%)
T1	20	(17.39%)	25	(25.73%)
T2	28	(24.34%)	53	(46.08%)
T3	58	(50.43%)	29	(25.21%)
T4	8	(6.95%)	7	(6.08%)
Gross type Fungating	20		(17.39%)	
Infiltrative	83		(72.17%)	
Ulcerofungating	3		(2.6%)	
Sessile	4		(3.47%)	
Solid	2		(1.73%)	
Tumor size	Ranged 0.6–8.0 cm		Mean 3.2 ± 1.6cm	
<4.5 cm	88		(76.53%)	
≥4.5 cm	27		(23.47%)	
N stage N0	66		(57.40%)	
N1	49		(42.60%)	
N2	0		(0%)	
M stage M0	107		(93.04%)	
M1	8		(6.96%)	
	Absent		Present	
Lymphatic invasion	64	(55.65%)	51	(44.35%)
Vascular invasion	101	(87.82%)	14	(12.18%)
Perineural invasion	48	(41.73%)	67	(58.26%)
Positive radial resected margin	107	(93.04%)	8	(6.96%)
Tumor ulceration	106	(92.17%)	9	(7.83%)
Histologic grade				
Well-differentiated	30		(26.08%)	
Moderately-differentiated	81		(70.43%)	
Poorly-differentiated	4		(3.47%)	
Histologic subtype				
Pancreaticobiliary subtype	30		(26.08%)	
Prone to pancreaticobiliary subtype	54		(46.95%)	
Prone to intestinal subtype	16		(13.91%)	
Intestinal subtype	14		(12.17%)	
Degree of fibrosis				
Absent	4		(3.47%)	
Mild	20		(17.39%)	
Moderate	64		(55.65%)	
Severe	27		(23.47%)	
Degree of inflammation				
Mild	39		(33.91%)	
Moderate	63		(54.78%)	
Severe	13		(11.30%)	
Tumor recurrence	Absent		Present	
	56	(48.69%)	52	(45.21%)
Death	37	(32.17%)	78	(67.82.%)
Follow-up duration (days)	Ranged 3–5234		969.7 ± 1135	
Disease free survival (days)	Ranged 3–4173		731.2 ± 954.5	

**Table 3 diagnostics-11-00597-t003:** Immunohistochemical staining conditions and the immunoreactivity of the immune checkpoint and EMT markers.

IHC Markers	PD-1		PD-L1				PD-L2				IGF-1		FGFR		VEGF	
			Score 1		Score 2		Score 1		Score 2							
Vendor	Cell Marque		Cell Marque				Cell Marque				abcam		abcam		Quartett Immuno diagnostika	
Dilution	1:100		1:1000				1:1000				1:100		1:100		1:50	
Positive control	Tonsil		Tonsil				Tonsil				Normal liver		Normal umbilical cord		Blood vessel	
	No.	(%)	No.	(%)	No.	(%)	No.	(%)	No.	(%)	No.	(%)	No.	(%)	No.	(%)
0							21	(18.4)	4	(3.5)	13	(11.3)				
1+	97	(85.1)	10	(8.8)	24	(21.1)	31	(27.2)	14	(12.3)	70	(60.9)	7	(6.1)	17	(14.8)
2+	11	(9.6)	52	(45.6)	44	(38.6)	39	(34.2)	34	(29.8)	26	(22.6)	50	(43.5)	71	(61.7)
3+	6	(5.3)	52	(45.6)	46	(40.3)	23	(20.2)	62	(54.4)	6	(5.2)	58	(50.4)	27	(23.5)
Total	114	(100)	114	(100)	114	(100)	114	(100)	114	(100)	115	(100)	115	(100)	115	(100)

## Data Availability

The data presented in this study are available on request from the corresponding author (https://www.researchgate.net/profile/Yosep-Chong. The data are not publicly available due to institutional policy.

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
