# Peer review of "High Expression of PD-L1 Is Associated with Better Survival in Pancreatic/Periampullary Cancers and Correlates with Epithelial to Mesenchymal Transition"

_diagnostics, 2021, doi:10.3390/diagnostics11040597_

Round 1

Reviewer 1 Report

Dear Authors

The paper of Nishant Thakur et al. evaluated the over expression of PD-L1 as a prognostic factor in pancreatic cancer associated with a better overall and disease free survival.

The topic is very interesting but the study design and the presentation of the results should be improved. The Authors based their work on comparing clinicopathological data, overall survival, disease free survival and the expression of immune checkpoint markers. This study includes patients who underwent pancreatectomy between 1998 and 2014. This inevitably leads to a selection bias because, in the last years, advances in management, chemotherapy and best supportive care, greatly improved the prognosis of these patients.

Furthermore, the analysis of the different clinicopathological parameters does not take into account several prognostic factors that are known to influence survival (e.g. staging, performance status, comorbidities, vascular resection, CA 19.9, adjuvant poly chemotherapies...). These aspects should be further investigated.  In the manuscript, there are also some inaccuracies (line 222) and typing errors (line 193-196). 

I strongly recommend authors to consider a re-submission after major revision. 

Author Response

Reviewer 1

The paper of Nishant Thakur et al. evaluated the over expression of PD-L1 as a prognostic factor in pancreatic cancer associated with a better overall and disease free survival.

Question 1. The topic is very interesting but the study design and the presentation of the results should be improved. The Authors based their work on comparing clinicopathological data, overall survival, disease free survival and the expression of immune checkpoint markers. This study includes patients who underwent pancreatectomy between 1998 and 2014. This inevitably leads to a selection bias because, in the last years, advances in management, chemotherapy and best supportive care, greatly improved the prognosis of these patients.

Answer 1.  Thank you for your comment. We clearly understood your point. We tried to include only cases with enough follow-up time and we thought exactly opposite about the recent advances in management, chemotherapy, and supportive care that these would rather alter the prognostic data of the recently treated patients more seriously in a better way and these might be cofounding factors for the survival analysis to evaluate the prostostic significance of PD-L1 expression. Furthermore, since the we authors have recently moved to the new hospital (Yeouido to Uijeongbu St. Mary’s Hospital), we sincerely ask for the reviewers’ kind understanding that we are not able to do additional study right now. We also have a IRB issue, that the materials after 2014 was not available without patients consents according to the Genetic Testing Law in Korer which can potentially lead a rather irregular patient recruitment. Instead of major change of study design, we added some explanation about these limitations of this study in the discussion as below.

Third, in this study, we only included cases with enough follow-up time and tried not to include too recent cases because recent advances in patient management, chemotherapy, and supportive quality might affect the overall and disease free survival of recent cases as cofounding factor other than the PD-L1 expression. However, this might not represent up-to-date results regarding many aspects. Thus, interpretation with caution is required.

Question 2. Furthermore, the analysis of the different clinicopathological parameters does not take into account several prognostic factors that are known to influence survival (e.g. staging, performance status, comorbidities, vascular resection, CA 19.9, adjuvant poly chemotherapies...). These aspects should be further investigated.

Answer 2.   Thank you for your valuable opinions. We also agree that this is a study from more of pathologic perspective. We analyzed the prognostic significance of T, N, M stage instead of staging because staging system for pancreatic cancer, ampulla of Vater, and common bile duct are all different (which also represents the prognostic significance of staging)

Most of the patients were operable cases which means performance status and comorbidities were not significantly varying between patients (stage IV cases were excluded because they were not eligible for operation and their performance status and comorbidities were fairly higher than other stage groups). We added a sentence about this in the methods as below.

…  Their performance status was fairly good for operation.  …

Cases with vascular resection was not many and the vascular resection was decided by the clinical situation in the operation room by case, thus it was not considered to take account seriously.

CA19.9 level was not suitable to be compared because the day of pre- or post operation testing of CA19-9 varied by patients. Some patients took the preoperational CA19-9 testing few months before the operation and some never had clear value because they took the tests outside the hospital during their health check-ups. Thus, we considered it is irrelevant to compare.

There was no patients with adjuvant chemotherapy and we mentioned in the original manuscript of the methods (page 2, 4.1. patient enrollment  None of the patients had experienced any preoperative chemotherapy treatment. page).

The problem we were focusing on this study was the prognostic significance of PD-L1 expression while the prognostic significance of other clinicopathological parameters are generally similar with previous findings (which indirectly proves our cohort is generally similar with other pancreatic cancer cohort). In this study, other parameter’s prognostic significance were exactly the same as generally known findings of the previous studies.

Question 3. In the manuscript, there are also some inaccuracies (line 222) and typing errors (line 193-196).

Answer 3. Thank you for your observation. We are very sorry for this editing error. We deleted the error.

Reviewer 2 Report

Interesting to read. High Quality of presentation. For better understanding, a short Explanation of the gross types would be helpful.

Author Response

Reviewer 2

Question 1. Interesting to read. High Quality of presentation. For better understanding, a short Explanation of the gross types would be helpful.

Answer 1. Thank you for your opinion. We added some more explanation about gross type in the methods as per your suggestion.

The gross type was classified based on the morphologic growth patterns of the tumours on the gross examination as fungating/polypoid, sessile, ulceroinfiltrative, and solid subtypes.

Reviewer 3 Report

  1. Automated image analyser should be a pioneer method to evaluate the expression level of PD-1, PD-L1 and PD-L2 in solid tumor. The immune system is very dynamic because lymphocyte, macrophage, and tumor cells has never been in quiesced state in solid tumor. This automated image analyser will be able to be applied to evaluate the dynamic status of immune cells, I think it is more promising tool in the future.
  2. CPS(Combined positive score) has been more used for prediction of immune-checkpoint inhibitor efficacy in solid tumors including gastric cancer and head and neck cancer.  Lymphocyte, macrophage and tumor cells express PD-L1, PD-1 and PD-L2 on the cell surface.  In this point of view, only evaluation of PD-1, PD-L1 and PD-L2 expression, we could know which cells express these molecules on their surface. These factors might be influence on the  conflicting results. 
  3. New scoring system in this article is a little bit complicated to use in the clinical settings. I think this scoring system could be modified more concise.

Author Response

Reviewer 3

Question 1. Automated image analyser should be a pioneer method to evaluate the expression level of PD-1, PD-L1 and PD-L2 in solid tumor. The immune system is very dynamic because lymphocyte, macrophage, and tumor cells has never been in quiesced state in solid tumor. This automated image analyser will be able to be applied to evaluate the dynamic status of immune cells, I think it is more promising tool in the future.

Answer 1. We agree with the reviewer opionion! Thank you for your comment.

Question 2. CPS(Combined positive score) has been more used for prediction of immune-checkpoint inhibitor efficacy in solid tumors including gastric cancer and head and neck cancer.  Lymphocyte, macrophage and tumor cells express PD-L1, PD-1 and PD-L2 on the cell surface.  In this point of view, only evaluation of PD-1, PD-L1 and PD-L2 expression, we could know which cells express these molecules on their surface. These factors might be influence on the conflicting results.

Answer 2. Thank you for your valuable comment.  We added your opinions in the discussion as below.

These conflicting results might also be from the complexity of tumour microenvi-ronment where tumour cells, infiltrating lymphocytes, and accompanying macrophages are all admixed. All of these cells can express PD-1, PD-L1, and PD-L2. However, our interpretation criteria, which is mostly combined positive score, does not properly rep-resent the different expression pattern of each group of cells.

Question 3. New scoring system in this article is a little bit complicated to use in the clinical settings. I think this scoring system could be modified more concise.

Answer 3. Thank you for your comment. We couldn’t agree with you more. We also wanted to make this system simple as possible easy to apply and remember. However, with the results of each category’s intensitity and proportion, we analyzed and drew out the most statistically significant cut-off values system so the system became complex as it dealt with numbers from 0~100%. Moreover, we guess image analysis system can also vary widely according to the systems, vendors, and pathology laboratories. Although it generally produces reproducible consistent analytic results within a certain lab within one examiner, it can also vary between different laboratories that have very different IHC quality and staining conditions. Thus, in this study, we simply tried to provide scientific evidence that higher expression of PD-L1 is significantly related to better OS and DFS in pancreatic cancers and immune check point markers are significantly associated with EMT markers. The scoring system should not be understood as exact value and we believe it surely needs external validation with larger sample size as we mentioned  in the discussion.

Round 2

Reviewer 1 Report

Dear Authors,

The topic provides a new and interesting perspective on pancreatic cancer. However, I have previously suggested considering a re-submission after major revision.

The new version still has some structural problems that affect its scientific quality. The reviews performed are not adequate in my opinion.

Several prognostic factors, well known to influence survival, are not investigated, others have been superficially analyzed. The performance status ("..fairly good for operation.."), vascular resection ("..it was not considered to take account seriously.."), adjuvant poly chemotherapies ("..There was no patients with adjuvant chemotherapy and we mentioned in the original manuscript of the methods (page 2, 4.1. patient enrollment  None of the patients had experienced any preoperative chemotherapy treatment.)").

Furthermore, in the revised manuscript, the authors claim they only included cases with enough follow-up time even if is also reported a follow-up period of 3 days (table 2).

All these aspects lead to misleading survival analysis and compromise the search result. If the primary endpoint of this study was to assess the prognostic significance of PD-L1 expression, I would recommend focusing on a more homogeneous and detailed cohort of patients.

Author Response

Reviewer 1

The topic provides a new and interesting perspective on pancreatic cancer. However, I have previously suggested considering a re-submission after major revision.

The new version still has some structural problems that affect its scientific quality. The reviews performed are not adequate in my opinion.

Question 1. Several prognostic factors, well known to influence survival, are not investigated, others have been superficially analyzed. The performance status ("fairly good for operation"), vascular resection (“.it was not considered to take account seriously."), adjuvant poly chemotherapies (“. There were no patients with adjuvant chemotherapy and we mentioned in the original manuscript of the methods (page 2, 4.1. patient enrollment  None of the patients had experienced any preoperative chemotherapy treatment.)").

Answer 1: Thank you for your comments.

We reviewed patients’ medical records once again and retrieved the American society of Anesthesiologists(ASA) Physical Status of all the patients. There were 10 patients with score 1, 103 patients with score 2, and only 2 patients with score 3. We added this in the methods, results and Table 2 as below.

Their performance status was checked according to American Society of Anaesthesiologists (ASA) for operation.

In ASA status score, 10 patients (8.69%) were found in score 1, 103 patients were in score 2, and only 2 patients were in score 3.

  1. We also reviewed all the patients’ record for vascular resection but none of the patients underwent vascular resection during the surgery. We added this in the methods as below.

    None of the patients underwent vascular resection during the surgery.

  2. We also checked chemotherapy status as you indicated, and it turned out that the patients over T stage 3 (>pT3) or nodal metastasis (>pN1) received 5FU or gemcitabine-based adjuvant chemotherapy. We revised manuscript as below. Sorry for not paying enough attention for your request last time.

    The patients over T stage 3 (>pT3) or nodal metastasis (>pN1) received 5FU or gemcitabine-based adjuvant chemotherapy.

Question 2. Furthermore, in the revised manuscript, the authors claim they only included cases with enough follow-up time even if is also reported a follow-up period of 3 days (table 2).

Answer 2:  Thank you for your comment. It is true that we tried to include patients with enough follow-up time. And the case you mentioned is one of the cases with acute post-operational complication such as hemorrhage or septic shock that is not relevant to recurrence of pancreatic cancer. So 3-day of follow up time for this patient is relevant and scientifically not problematic and it does not represent ‘not enough follow-up time’.

Question 3. All these aspects lead to misleading survival analysis and compromise the search result. If the primary endpoint of this study was to assess the prognostic significance of PD-L1 expression, I would recommend focusing on a more homogeneous and detailed cohort of patients.

Answer 3: Thank you for your comment. During the designing of this study, we have reviewed several studies with similar study design (shown below). These studies included the age, sex, pathological tumor size, lymph node involvement, pathological type, tumor grade, molecular subtype, DFS, location, T stage, N stage, and distant metastasis, etc., as our study. However, none of these studies included the vascular resection, performance status and chemotherapy. So we did not pay enough attention to these parameters. We hope newly added information would help clarifying the homogeneity of the cohort in our study. Thank you.

  1. Prognostic value of PDL1 expression in pancreatic cancer https://dx.doi.org/10.18632%2Foncotarget.11685

In this study the author used 453 clinical pancreatic cancer samples and the age, sex, pathological tumor size, pathological lymph node, pathological type, tumor grade, surgery, baileys molecular subtype and median DFS were the clinicopathological factors.   

  1. The coexpression and clinical significance of costimulatory molecules B7-H1, B7-H3, and B7-H4 in human pancreatic cancer https://dx.doi.org/10.2147%2FOTT.S66809

Here the author exploited 63 pancreatic cancer samples and the age, sex, Location, tumor size, tumor grade, tumor status, nodal status, and distant metastasis were the clinicopathological factors

Reviewer 2 Report

Comprehensive.

Author Response

Thank you for your acceptance.
